# A live imaging system to analyze spatiotemporal dynamics of RNA polymerase II modification in *Arabidopsis thaliana*

Mio K. Shibuta[1], Takuya Sakamoto[2], Tamako Yamaoka[2], Mayu Yoshikawa[2], Shusuke Kasamatsu [3], Noriyoshi Yagi [2], Satoru Fujimoto[2], Takamasa Suzuki [4], Satoshi Uchino[5], Yuko Sato[5,6], Hiroshi Kimura [5,6] & Sachihiro Matsunaga [1✉]

Spatiotemporal changes in general transcription levels play a vital role in the dynamic regulation of various critical activities. Phosphorylation levels at Ser2 in heptad repeats within the C-terminal domain of RNA polymerase II, representing the elongation form, is an indicator of transcription. However, rapid transcriptional changes during tissue development and cellular phenomena are difficult to capture in living organisms. We introduced a genetically encoded system termed modification-specific intracellular antibody (mintbody) into *Arabidopsis thaliana*. We developed a protein processing- and 2A peptide-mediated two-component system for real-time quantitative measurement of endogenous modification level. This system enables quantitative tracking of the spatiotemporal dynamics of transcription. Using this method, we observed that the transcription level varies among tissues in the root and changes dynamically during the mitotic phase. The approach is effective for achieving live visualization of the transcription level in a single cell and facilitates an improved understanding of spatiotemporal transcription dynamics.

[1] Graduate School of Frontier Sciences, Department of Integrated Biosciences, The University of Tokyo, Kashiwa, Chiba, Japan. [2] Faculty of Science and Technology, Department of Applied Biological Science, Tokyo University of Science, Noda, Chiba, Japan. [3] Academic Assembly (Faculty of Science), Yamagata University, Yamagata, Japan. [4] College of Bioscience and Biotechnology, Chubu University, Kasugai, Aichi, Japan. [5] Graduate School of Bioscience and Biotechnology, Tokyo Institute of Technology, Midori-Ku, Yokohama, Japan. [6] Cell Biology Center, Institute of Innovative Research, Tokyo Institute of Technology, Midori-Ku, Yokohama, Japan. ✉email: sachi@edu.k.u-tokyo.ac.jp

Dynamic changes of post-translational modifications, such as histone modifications, DNA methylation, and RNA polymerase II (RNAPII) modifications, play critical biological functions in fundamental cellular processes. The C-terminal domain (CTD) of the large subunit of RNAPII contains highly conserved heptad repeats, which are subjected to several post-translational modifications in the transcription cycle. Phosphorylation of CTD Ser5 (Ser5P) and Ser2 (Ser2P) residues acts as an essential factor in the transition of transcription cycles. The Ser5P residue is required to facilitate the transition between transcription initiation and transcription elongation, and Ser2P signifies transcription termination[1–3]. In *Arabidopsis thaliana*, global run-on sequencing (GRO-seq) and plant native elongating transcript sequencing (pNET-seq) revealed that RNAPII modification levels of Ser5P and Ser2P accumulate along the gene body of active genes, and especially Ser2P accumulation shows sharp peaks around the polyadenylation site[3–5]. In analyses of these modifications, immunostaining based on fluorescence microscopy has been widely used. Immunostaining revealed that Ser2P and Ser5P of RNAPII are enriched within transcriptionally active euchromatin and are absent from nucleoli and heterochromatin[6,7]. Using super-resolution microscopic techniques, such as structured illumination microscopy or photoactivated localization microscopy, the localization of Ser2P or Ser5P in fixed cell nuclei can be resolved at the single-molecule level[6,7]. However, several issues still need to be resolved for spatiotemporal resolution and quantification of modification levels in living cells. It is impossible to monitor the dynamics of RNAPII modification during cellular processes because fixation is an inevitable step in immunofluorescence observation.

Recently, live-cell imaging systems for observing post-translational modifications have been established. For example, genetically encoded proteins that contain domains or proteins with affinity for methylated DNA and fluorescent proteins can be used for real-time monitoring of CG and CHH methylation in mammals and *A. thaliana*[8,9]. Modifications of RNAPII and histones also can be monitored by two methods based on the use of modification-specific antibodies: fluorescent-labeled antigen-binding fragment-based live endogenous modification labeling (Fab-LEM) and modification-specific intracellular antibodies (mintbodies)[10]. In Fab-LEM, Fabs are loaded into cells by injection or a beads-loading method[10–12]. Mintbodies are genetically encoded probes with a single-chain variable fragment (scFv) fused to a fluorescent protein and can be expressed in vivo[10,13–15]. Both methods enable the distribution and levels of modifications in living cells to be monitored without disturbing growth and development because both probes bind to targets only for several seconds without blocking access to common target modifications[10,13]. The mintbody without injection or a beads-loading method is advantageous in tracking spatiotemporal activity in plant tissues because it can be expressed in any cell at any time point under the control of relevant promoters.

In this study, we developed a genetically encoded fluorescent probe for RNAPII Ser2P using a mintbody in *A. thaliana* for live-cell imaging. We cloned cDNA from the 42B3 hybridoma cell line encoding the variable regions of heavy ($V_H$) and light ($V_L$) chains in the scFv[12,16]. To quantify the modification level, we introduced a two-component system with 2A peptide-mediated co-expression to evaluate modification levels through live-cell imaging. The two-component system greatly aids in quantifying the fluorescence intensity of the mintbody and enables quantitative tracking of transcription dynamics. We observed that the modification levels showed tissue-specific variation in root tissues. Time-lapse imaging during the mitotic (M) phase revealed that the modification level rapidly decreased when chromatin condensation began, showing intense transcription repression. The Ser2P-mintbody provided spatiotemporal information in *Arabidopsis* seedlings and allowed analysis of nuclear distributions and modification levels in real time during various cellular functions.

## Results

**Introduction of the Ser2P-specific mintbody into *A. thaliana* seedlings.** To generate a mintbody that can specifically bind to Ser2P, we cloned cDNA from the 42B3 hybridoma cell line for use as the scFv coding sequence. We constructed an expression vector by fusing the scFv to *EGFP* at the C-terminus under control of the *UBQ10* promoter (Fig. 1a). In the transgenic seedlings, Ser2P-mintbody induced no apparent phenotypic alteration (Supplementary Fig. 1), indicating that Ser2P-mintbody is not disruptive of plant growth.

To validate the function of Ser2P-mintbody in *A. thaliana* seedlings, we checked the nuclear distribution of Ser2P-mintbody by immunostaining. Ser2P-mintbody showed a similar distribution pattern to that of Ser2P and was excluded from heterochromatin (Fig. 1b). Note that the anti-GFP marker signal would include free Ser2P-mintbody that is not bound to Ser2P at the moment of immunostaining, and that anti-GFP antibody for Ser2P-mintbody and the anti-Ser2P antibody would compete for the same target, which might be reflected in the minor difference between the anti-GFP signal and the anti-Ser2P signal (Fig. 1b).

To further assess Ser2P-mintbody specificity, we performed chromatin immunoprecipitation followed by sequencing (ChIP-seq) using *A. thaliana* seedlings. This analysis showed that Ser2P-mintbody was enriched, especially after transcriptional end sites (TES) of representative transcriptionally active genes, such as *ACT2* and *FAS1*, and was not enriched for a representative transcriptionally repressed gene (*FLC* in Col) or a silenced transposon (*ATCOPIA41*), which was similar to the Ser2P plot obtained using the anti-Ser2P antibody (Fig. 1c). The average profile plots showed that the distribution of Ser2P-mintbody and Ser2P around their peaks was similar (Fig. 1d, e, and Supplementary Fig. 2a). The peak around TES was clearly not an artifact from the comparison with the negative control (EGFP-3xFLAG; Supplementary Fig. 2b). A large proportion of the target genes of Ser2P-mintbody overlapped with those of Ser2P (Fig. 1f and Supplementary Fig. 2c). The number of target genes of Ser2P-mintbody was smaller compared with that of Ser2P (Fig. 1f and Supplementary Fig. 2c). However, when replotted for common targets, anti-Ser2P antibody-specific targets, and anti-GFP antibody-specific targets, respectively, each group showed similar enrichment patterns, suggesting that Ser2P-mintbody could bind to the majority of Ser2P target genes to a certain degree (Supplementary Fig. 2d–g). We note, in passing, that Ser2P profiles reported previously using ChIP-seq did not show clear peaks around the TES, as reported by more sophisticated methods such as GRO-seq and pNET-seq[3,17]. The reason that we were able to obtain such clear peaks using ChIP-seq may be due to our sonication protocol as detailed in the "Materials and methods" section. In any case, these data provided strong evidence that Ser2P-mintbody specifically bound to Ser2P in living plants.

**Time-lapse quantification of Ser2P modification.** Previous reports showed that mintbodies are reversibly mobile between the cytoplasm and nucleus depending on the target modification levels. Thus, the nuclear-to-cytoplasmic ratio of fluorescence intensity of the mintbody was calculated to quantify the modification levels in cultured cells or yeast cells[13–15]. However, this method is not suitable for *A. thaliana* seedlings; *A. thaliana* cell structures are more complicated than isolated cells, and the cytoplasm often contains many, sometimes large vacuoles. Given

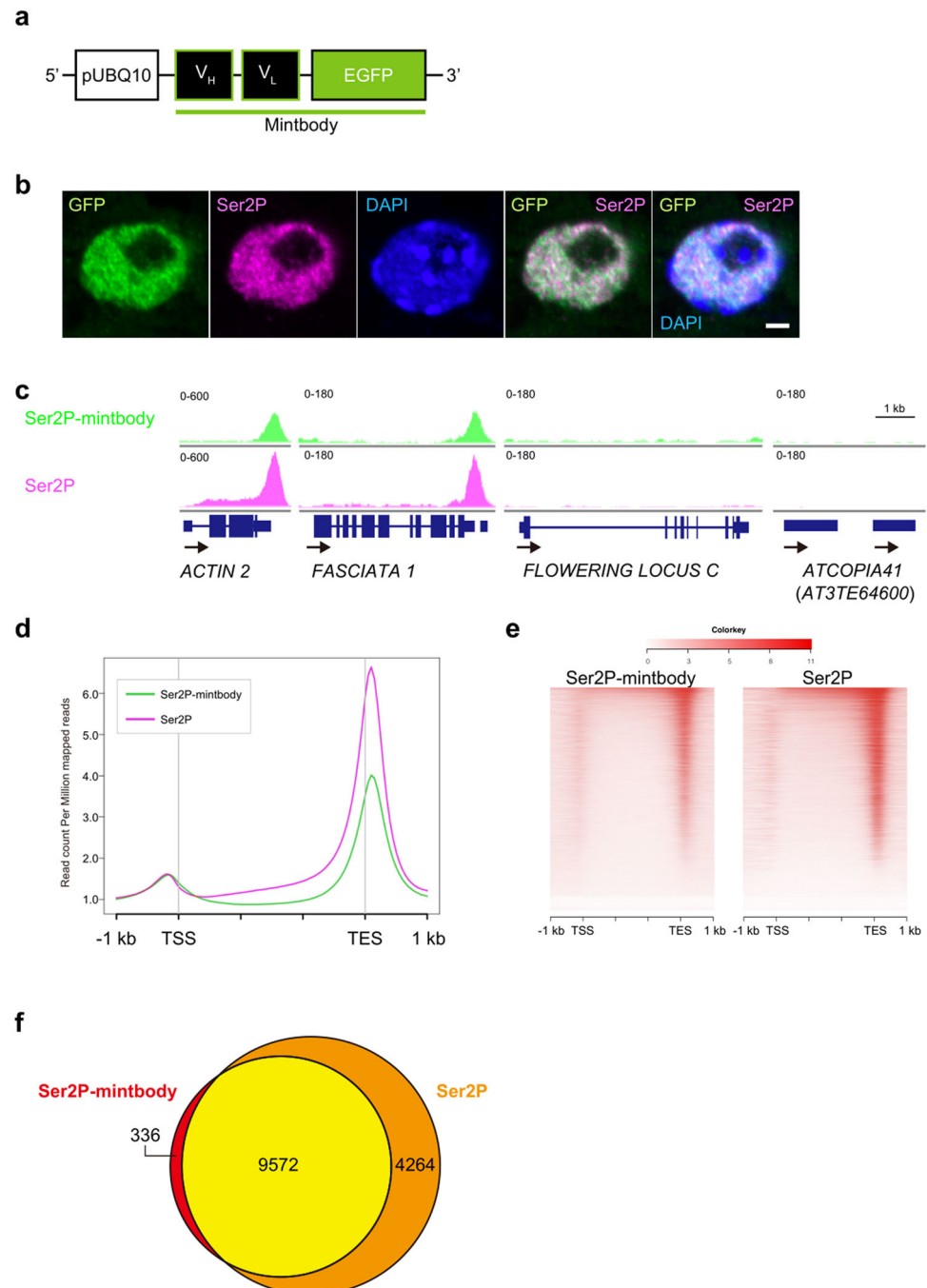

**Fig. 1 Capacity of Ser2P-mintbody to bind with Ser2P in living *Arabidopsis* seedlings. a** Schematic diagram of the expression cassette of Ser2P-mintbody. pUBQ10: *UBIQUITIN10* (AT4G05320) promoter; V$_H$: heavy chain variable domain of anti-Ser2P antibody; V$_L$: light chain variable domain of anti-Ser2P antibody. **b** Immunostained images of Ser2P-mintbody and Ser2P. Nuclei in root expressing *pUBQ10:Ser2P-mintbody* were fixed, immunostained with anti-GFP (green) and anti-Ser2P (magenta) antibodies, and stained with DAPI (blue). Scale bar: 2 μm. **c** Genome-browser view of ChIP-seq peaks at representative loci: active genes, *ACTIN 2* (AT3G18780) and *FASCIATA 1* (AT1G65470); a repressed gene, *FLOWERING LOCUS C* (AT5G10140); a transposon, *ATCOPIA41* (AT3TE64600). **d, e** Distribution of Ser2P-mintbody and Ser2P vs. gene-body shown as an average profile plot (**d**) and as heatmaps (**e**). TSS transcription start site, TES transcription end site, −1 kb: 1 kb upstream of TSS; 1 kb: 1 kb downstream of TES. **f** Venn diagram showing the overlap of Ser2P-mintbody- and Ser2P-enriched genes.

these characteristics, unambiguous measurement of cytoplasmic fluorescent intensity is difficult. Thus, to quantify modification levels of Ser2P in *A. thaliana* seedlings, we proposed an evaluation method based on a "two-component system": Ser2P-mintbody and a standard protein were expressed simultaneously in the same amount, and the modification level was calculated as the ratio of the nuclear fluorescence intensity of

Ser2P-mintbody to that of the standard protein (Fig. 2a). To generate the two-component system, we focused on translation-coupled protein processing using the IntF2A-based co-expression system[18,19] (Fig. 2b). This system enables the coordinated expression of two proteins from a single transgene through co-translational cleavage caused by 2A peptide's translational recoding activity followed by post-translational autocatalytic

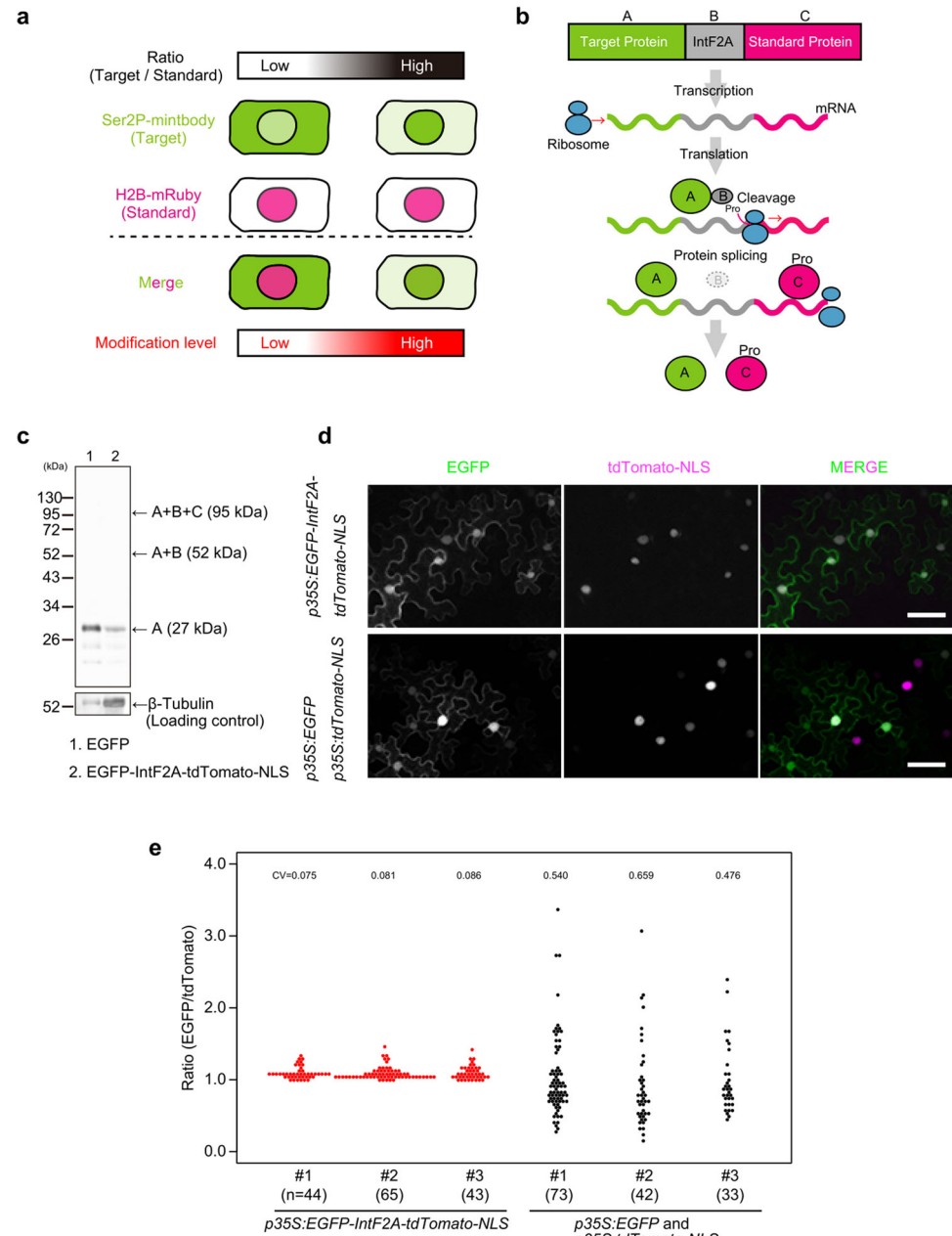

**Fig. 2 Intein and 2A peptide-mediated co-expression for the two-component system. a** Schematic diagram of the strategy of the two-component system. The target protein (Ser2P-mintbody) with a fluorescence protein and the standard protein (H2B-mRuby) with another fluorescence protein are co-expressed in cells of interest, and the ratio of fluorescence intensity of the target protein to that of the standard protein in nuclei is measured. As Ser2P-mintbody transiently binds to Ser2P and freely passes through nuclear pores, the ratio should reflect the Ser2P modification level in each cell. **b** Schematic diagram of the IntF2A-based co-expression system. The expression cassette is assembled by connecting an upstream target protein (molecule A) and a downstream standard protein (molecule C) with the IntF2A auto-processing domain (molecule B). Co-translational cleavage via the 2A-peptide produces the target protein with a remnant IntF2A residue and the standard protein with a proline residue (Pro), and then post-translational auto-cleavage via intein releases the target protein. As a result, the target protein and the standard protein are co-translated from the single mRNA in the same proportion in all cells. **c** Efficient auto-cleavage and release of the target protein in *Nicotiana benthamiana*. In western blotting analysis, the total protein extract was probed with anti-GFP antibody for detection of the released target protein. A + B + C, A + B, and A denote unprocessed protein, the target protein with IntF2A, and totally processed protein, respectively. β-tubulin was used as a loading control. For the full-length western blots see Supplementary Fig. 4. **d** Transient expression with the two-component system in *N. benthamiana*. Epidermal cells in leaves were observed 3 days after infiltration of *p35S:EGFP-IntF2A-tdTomato-NLS* or a set of *p35S:EGFP* and *p35S:tdTomato-NLS*. EGFP and tdTomato fluorescence is green and magenta, respectively. Scale bars, 50 µm. **e** Ratios of EGFP fluorescence intensity relative to tdTomato fluorescence intensity in *N. benthamiana*. The epidermal tissue in leaves was observed 3 days after infiltration of *p35S:EGFP-IntF2A-tdTomato-NLS* or a set of *p35S:EGFP* and *p35S:tdTomato-NLS*. Ratios were calculated from 33 to 73 cells in each leaf, and values are shown in dot plots. n=biologically independent cells. CV coefficient of variation. Source data underlying the plot are available as Supplementary Data 1.

cleavage through intein at its N-terminal junction. IntF2A includes a 58 amino acid F2A sequence that shows high cleavage efficiency and a mutated intein sequence that can cleave the protein flanking the intein's N-terminus. The system does not require any host-specific factors, thus it should be applicable in a wide range of hosts.

First, to examine if the system is suitable for the proposed evaluation method, we prepared IntF2A-based co-expression cassettes, which were assembled by connecting the upstream target protein (EGFP) and the downstream standard protein (tdTomato-NLS) with the IntF2A auto-processing domain that enables self-excision at both terminal junctions. We then checked the self-excision efficiency and ratio of the target protein to the standard protein in a transient expression assay in *N. benthamiana* leaves. The complete release of the processed EGFP was confirmed from the migration to the same position as the control (about 27 kDa), suggesting that 2A peptide-mediated cleavage and subsequent intein-mediated cleavage functioned as expected (Fig. 2c and Supplementary Fig. 4a, b). Next, we calculated the ratio of the integrated nuclear fluorescence intensity of EGFP to that of tdTomato. The dispersion of the ratio in the two-component system was much smaller than that in the system employing co-transformation of EGFP and tdTomato from different plasmids (Fig. 2d, e). This result indicated that the target and standard proteins were expressed in equal ratios across all cells.

Next, we constructed a cassette containing Ser2P-mintbody as the target protein and H2B-mRuby as the standard protein for spatiotemporal observation in *A. thaliana* seedlings (Fig. 3a). Given that the same ribosome can translate both Ser2P-mintbody and H2B-mRuby from the same mRNA, the ratio of the proteins expressed in a cell should be equal across all cell types. Ser2P-mintbody can reversibly migrate between the nucleus and cytoplasm depending on the modification level, whereas H2B-mRuby is stably localized in the nucleus. Therefore, the modification levels of Ser2P could be measured as the ratio between the intensities of Ser2P-mintbody and H2B-mRuby in the nucleus. The majority of seedlings showed a similar ratio distribution (Supplementary Fig. 3a, b), suggesting that the two-component system could act as a precise evaluation method in *A. thaliana* seedlings. To assess if this line could reflect rapid changes in Ser2P level, we measured the modification levels under drug-induced hypophosphorylated conditions. Flavopiridol (FP) blocks the kinase activity of positive transcription elongation factor b and FP treatment reduces the level of Ser2P in *A. thaliana* seedings[20,21]. Western blotting showed that the Ser2P level decreased to be undetectable at 30 min (Fig. 3b and Supplementary Fig. 5a–d). Time-lapse imaging of the line showed that Ser2P-mintbody diffused away from the nucleus with time, and the ratio continued to decrease for 60 min after which the value stayed constant for the remainder of the measurement period (Fig. 3c, d). After FP removal by washing, the level of Ser2P recovered after 4 h (Fig. 3e and Supplementary Fig. 6a–d). Ser2P-mintbody also relocalized in the nucleus after washing, thus reaffirming the preferential distribution in the nucleus (Fig. 3f). The ratio decreased in response to FP treatment and gradually recovered after washing (Fig. 3g). These data indicated that the IntF2A-based two-component system was effective in quantifying the rapid spatiotemporal dynamics of Ser2P levels in living cells.

**Transcription levels varied depending on root layers**. *Arabidopsis thaliana* has highly specialized cell types, and individual genes, pathways, and metabolites exhibit diverse tissue-dependent behavior. The primary root shows a radial pattern of tissue types arranged in concentric layers[22]. To generate a tissue-specific map of transcriptional activity, we observed the root elongation zone and calculated the relative ratio of the nuclear fluorescence intensity of GFP to that of mRuby in the epidermis, cortex, endodermis, and pericycle in *pRPS5a:EGFP-IntF2A-H2B-mRuby* and *pRPS5a:Ser2P-mintbody-IntF2A-H2B-mRuby* seedlings (Fig. 4a–d). The dispersion of the ratio in each layer in *pRPS5a: EGFP-IntF2A-H2B-mRuby* seedlings was relatively fixed (CV = 0.011–0.024) (Fig. 4c). In *pRPS5a:Ser2P-mintbody-IntF2A-H2B-mRuby* seedlings, the dispersion of the ratio in the pericycle was much larger than those in other layers (Fig. 4d), showing cells are present in various transcription states in the pericycle.

To compare the transcription levels among root layers, we replotted values from *pRPS5a:Ser2P-mintbody-IntF2A-H2B-mRuby* seedlings divided by the median of the relative ratio in each layer in *pRPS5a:EGFP-IntF2A-H2B-mRuby* seedlings (Fig. 4e). The values in the endodermis were lower than those in the epidermis and pericycle, indicating that transcription activity in the root varied among tissues.

**Transcription activity was strongly reduced during the M phase**. In general, transcription is forcefully silenced during the M phase because the transcription complexes are inactivated and disassembled from chromatin during chromatin condensation in the prophase[23]. Recently, GRO-seq revealed the cell-cycle-specific distribution landscape of active RNAPII in a synchronized human cultured cell line[24].

To visualize the cellular dynamics of RNAPII Ser2P in plants, we performed time-lapse imaging during the M phase in the root meristematic zone of *pRPS5a:EGFP-intF2A-H2B-mRuby* and *pRPS5a:Ser2P-mintbody-intF2A-H2B-mRuby* seedlings. We obtained Z-stacked images at 5-min intervals. Both EGFP and Ser2P-mintbody rapidly diffused into the cytoplasm in prometaphase when chromatin condensation started, and gradually relocalized in the daughter nuclei from telophase when chromatin started to decondense (Fig. 5a, b). The relative intensity ratios of both EGFP and Ser2P-mintbody rapidly decreased in prometaphase, remained at a low level until anaphase, and then gradually recovered to the interphase (Fig. 5c). The ratio transition patterns were similar between EGFP and Ser2P-mintbody, suggesting that Ser2P-mintbody behaved similarly to EGFP because there were few Ser2P modifications during the M phase. These data strongly suggested that transcription was strongly repressed from the beginning of prometaphase, and gradually restarted coincident with chromatin decondensation in the daughter nuclei.

## Discussion

In this study, we incorporated the RNAPII Ser2P-mintbody within an IntF2A-mediated two-component system to monitor RNAPII Ser2P dynamics and modification levels in living *A. thaliana* seedlings. Our approach revealed spatial variation in transcription levels in root layers and the temporal dynamics of strong transcription repression during the M phase.

The mintbody can be used to visualize the spatial localization of RNAPII Ser2P at the nucleus level. In addition, we show that the Ser2P-mintbody construct can be used to map the enrichment of RNAPII Ser2P genome-wide in ChIP-seq analysis. The mintbody showed a lower peak and fewer target genes than those for the anti-Ser2P antibody, reflecting the moderate affinity of Ser2P-mintbody for the targets. This at least partially reflects its structure; mintbodies are monovalent, whereas normal antibodies are bivalent. Furthermore, the generated lines expressing Ser2P-mintbody in the present study grew normally without aberrant phenotypes (Supplementary Fig. 1).

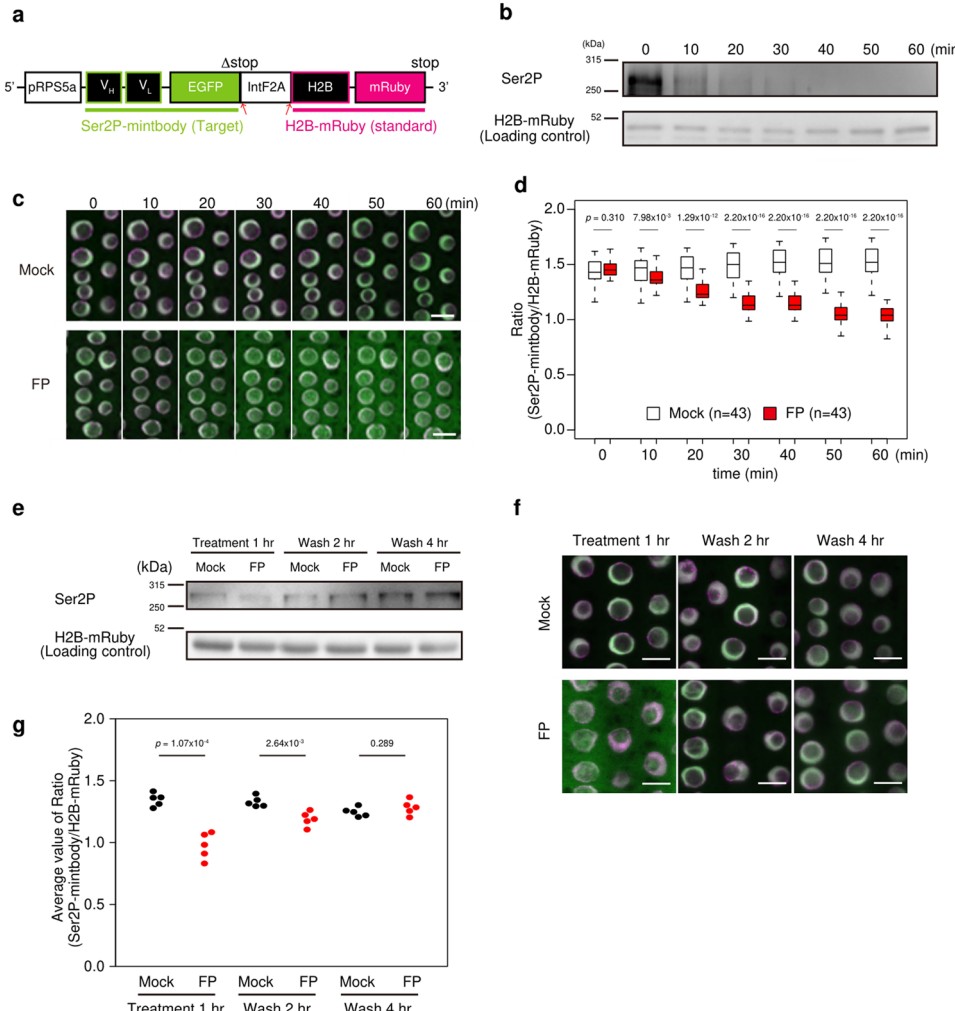

**Fig. 3 Quantification of Ser2P levels under chemically induced transcriptional repression. a** Schematic diagram of the Ser2P-mintbody and H2B-mRuby expression cassette. pRPS5a: *RIBOSOMAL PROTEIN 5A* (AT3G11940) promoter. Red arrows indicate cleavage and processing positions. **b** Western blotting analysis of dynamics of Ser2P under flavopiridol (FP) treatment. The total protein extract from whole seedlings was probed with anti-Ser2P antibody after FP addition. H2B-mRuby was used as a loading control. For the full-length western blots see Supplementary Fig. 5. **c** Time-lapse imaging under FP treatment. Seedlings were incubated in the presence or absence of FP for 1 h. Merged images of Ser2P-mintbody (green) and H2B-mRuby (magenta) in cortex cells in the root transition zone are shown. Images are sum slices of Z-stacks. Scale bars, 10 μm. **d** Time dependence of the ratio of Ser2P-mintbody fluorescence intensity relative to H2B-mRuby fluorescence intensity under mock or FP treatment. The ratios were measured from 43 cells at each time point in the root transition zone and are shown in box plots. *p* Values were calculated with the Wilcoxon rank sum two-sample test. *n* = biologically independent cells. Source data underlying the plot are available as Supplementary Data 2. **e** Western blotting analysis of dynamics of Ser2P level. After incubation in FP for 1 h, seedlings were washed and incubated in a standard medium, and the total protein extract from whole seedlings was probed with anti-Ser2P antibody at 1 h after FP addition and 2 h/4 h after washing. H2B-mRuby was used as a loading control. For the full-length western blots see Supplementary Fig. 6. **f** Live cell imaging under FP treatment. Merged images of Ser2P-mintbody (green) and H2B-mRuby (magenta) are shown. Scale bars, 10 μm. **g** Ratio of fluorescence intensity of Ser2P-mintbody relative to H2B-mRuby under FP and washing treatment. Ratios were calculated from more than 38 cells from each root at each time point, and average values from five roots are shown in dot plots. *p* Values were calculated with two-tailed unpaired two-sample *t* tests. *n* = biologically independent cells. Source data underlying the plot are available as Supplementary Data 3.

The present tissue-specific observations showed that the transcription level in pericycle cells was relatively variable compared with other tissues. Lateral roots originate from a subset of pericycle cells, so the variation might reflect the status of presumptive lateral root primordia and other cells. Therefore, it would be interesting to observe lateral root development with lateral root development reporter genes because transcription levels are predicted to change dramatically during this process[25–28]. In addition, the M-phase observations revealed that transcription was dynamically changed throughout the M phase. It is presumed that on/off switching of transcription is strongly involved with chromatin condensation during the M phase, so higher-resolution

observation of Ser2P-mintbody will contribute to understanding the processes of chromatin condensation and decondensation, especially the orientation points on chromatin.

Immunostaining and Ser2P-mintbody are practical tools for observation of Ser2P in *A. thaliana*. Immunostained samples showed a high fluorescence intensity and could be used for super-resolution microscopy, which allowed us to observe the distribution pattern of Ser2P and quantify the absolute number of phosphorylated RNAPII molecules with high precision[6,7]. However, with immunostaining, observation of dynamic changes in modification levels during development, differentiation, and stress responses is difficult. The Ser2P-mintbody in conjunction

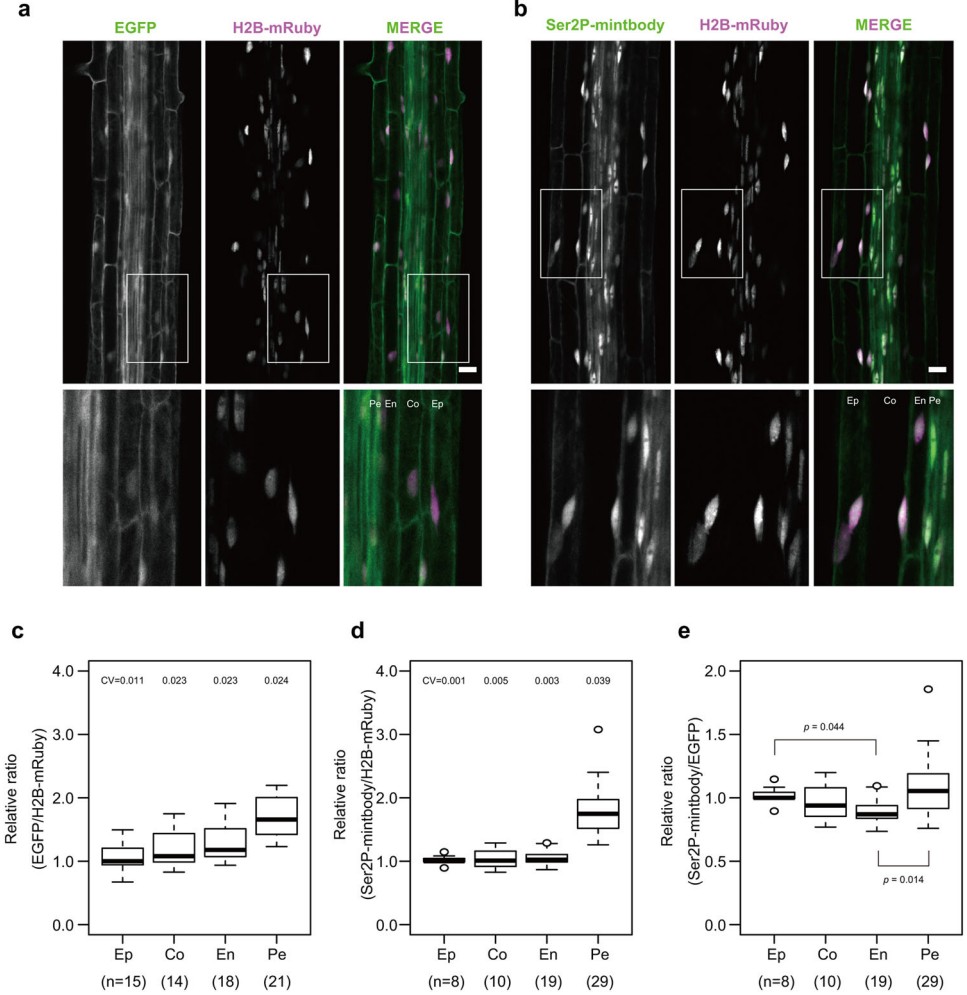

**Fig. 4 Variation of general transcription activity among tissues in the root. a**, **b** Live cell imaging of EGFP or Ser2P-mintbody and H2B-mRuby in the root elongation zone. Panels are sum slices of Z-stacks. Lower panels are enlarged from the inset of upper panels. Scale bars, 20 μm. Ep Epidermis, Co Cortex, En Endodermis, Pe Pericycle. **c**, **d** EGFP or Ser2P-mintbody/H2B-mRuby intensity ratios in epidermis, cortex, endodermis, and pericycle cells normalized by the median ratio in the epidermis. Data from three distinct roots (three distinct normalization scales) are accumulated. $n$ = biologically independent cells. CV coefficient of variation. **e** Ser2P levels in Fig. 4d divided by the median ratio in each tissue in Fig. 4c. $p$ Values were calculated with the Bonferroni-adjusted unpaired Wilcoxon rank sum test. $p$ Values less than 0.05 are shown. $n$ = biologically independent cells. Source data underlying the plots are available as Supplementary Data 4.

with the two-component system is undoubtedly an invaluable tool to solve these problems, as Ser2P-mintbody is advantageous for live imaging and quantification of the Ser2P modification level in intact seedlings. Therefore, it allows elucidation of spatio-temporal transcription dynamics in various biological processes in combination with specific reporter lines. Accordingly, for observation of post-translational Ser2 modifications, immunostaining or Ser2P-mintbody could be selected with consideration of their relative advantages and disadvantages.

In addition to imaging experiments, Ser2P-mintbody could be applied in other techniques. For example, Ser2P-mintbody could be used for tissue-specific gene expression analysis. Several methods are already established, such as direct tissue isolation[29] and isolation of nuclei tagged in specific cell types[30]. Tissue-specific gene expression analysis is increasingly essential to elucidate highly structured biological activities. Given that Ser2P-mintbody is effective as a functional endogenous antibody in *A. thaliana* seedlings for ChIP-seq analysis, tissue-specific ChIP-seq analysis could be conducted to visualize transcriptionally active genes if Ser2P-mintbody is expressed under the control of tissue-specific or phase-specific promoters. The present method serves

as a valuable tool for live visualization of the general transcription level and facilitates an improved understanding of the spatio-temporal transcription dynamics in plants.

## Methods

**Plant materials and growth conditions**. The *A. thaliana* accession Columbia-0 (Col) was used as the wild type. All transgenic lines used were in the Col background. Seeds were germinated on soil or on Murashige and Skoog (MS) medium supplemented with 0.8% agar and 1% sucrose. The seeds were stratified at 4 °C for 1 day, then incubated at 22 °C under long days (16 h light/8 h dark).

**Plasmid construction**. The plasmid pUBQ10:Ser2P-mintbody was constructed as follows. First, the destination vector pGWB-UBQ10 was generated by modification of pGWB501[31]. The 636 bp *UBQ10* promoter was inserted into *Hinc*II-digested pUC19; this plasmid was then digested by *Sbf*I and *Xba*I, and the resulting SbfI-UBQ10-XbaI fragment was inserted into *Sbf*I and *Xba*I-digested pGWB501. Next, the 42B3scFv coding sequence and GFP with the pENTR/D-TOPO vector sequence (the 2 x p35S:19E5scFv-GFP plasmid[15] was used as a template sequence) were amplified by PCR and fused to both ends of the fragments using the NEBuilder HiFi DNA Assembly Master Mix (New England BioLabs). Following this process, we detected a frame-shift mutation in the coding sequence, so we revised the plasmid using other primers; the construct was then recombined into the pGWB-UBQ10 vector using LR Clonase II (Thermo Fisher Scientific).

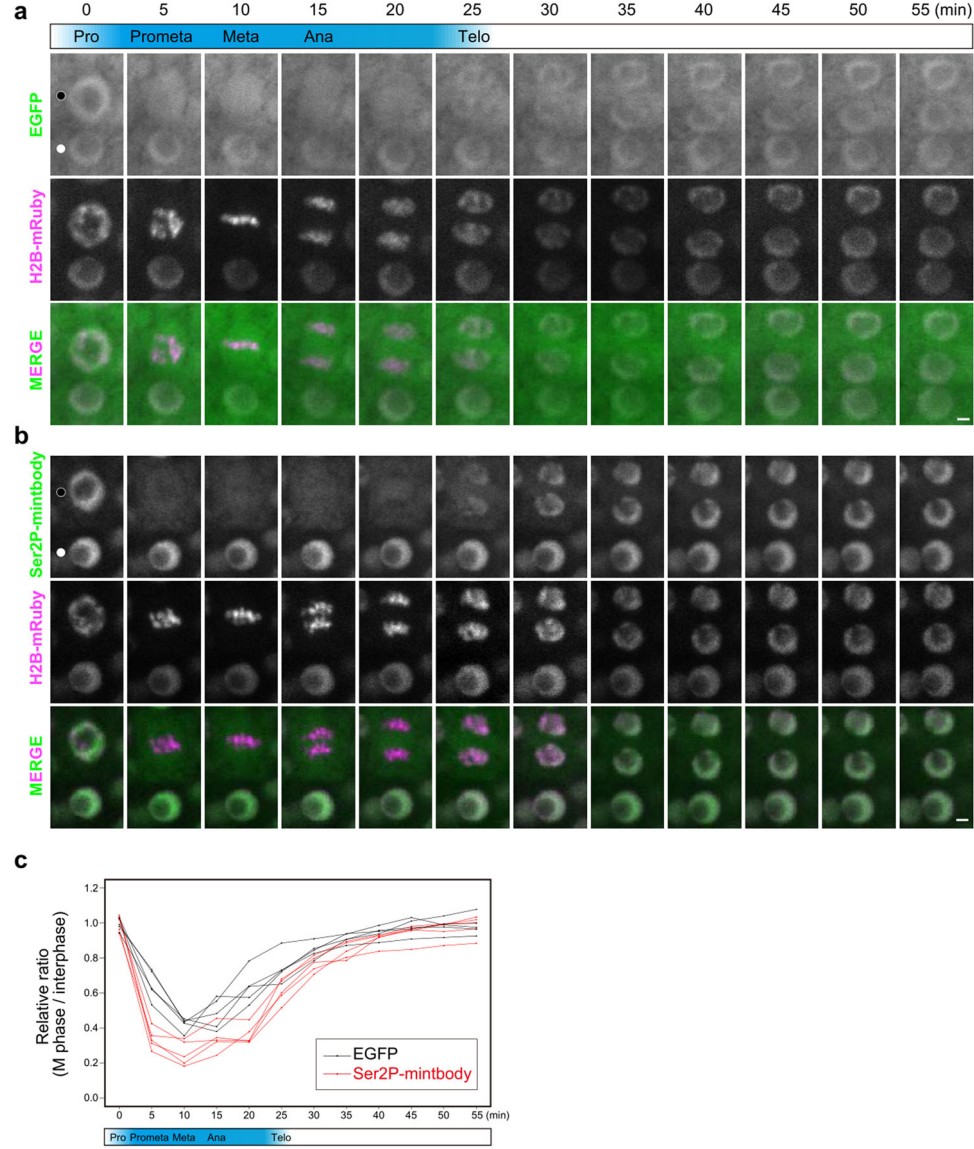

**Fig. 5 Dynamics of general transcription activity during the M phase. a**, **b** Time-lapse imaging of the set of EGFP and H2B-mRuby (**a**), and Ser2P-mintbody and H2B-mRuby (**b**) in cortex cells in the root meristematic zone. Representative images of M-phase nuclei (filled circles) and interphase nuclei (open circles) are shown. Images are sum slices of Z-stacks. Scale bars, 2 μm. **c** Dynamics of Ser2P level during the M phase. The EGFP or Ser2P-mintbody/H2B-mRuby intensity ratios in M-phase nuclei and those of interphase nuclei located near the M-phase nuclei were calculated and M phase/interphase ratios are plotted. Source data underlying the plot are available as Supplementary Data 5.

To construct p35S:EGFP-3xFLAG, the coding sequences for EGFP and 3xFLAG were amplified by PCR and recombined into the *Nde*I and *Sal*I-digested pRI201-AN vector (TaKaRa) using the NEBuilder HiFi DNA Assembly Master Mix.

To construct pRPS5a:EGFP-IntF2A-H2B-mRuby and pRPS5a:Ser2P-mintbody-IntF2A-H2B-mRuby, the cDNA sequences encoding HTB1 (AT1G07790), IntF2A (an artificially synthesized sequence was used as the template), and mRuby3 were amplified by PCR and recombined into the *Nde*I and *Sal*I-digested pRI201-AN vector using the NEBuilder HiFi DNA Assembly Master Mix. The 1665 bp pRPS5a sequence was amplified by PCR and subcloned into the *Hind*III and *Xba*I-digested pRI201 vector. The coding sequences for EGFP or Ser2P-mintbody and IntF2A-H2B-mRuby were amplified by PCR and assembled into the *Nde*I and *Sal*I-digested pRI201-pRPS5a vector using the NEBuilder HiFi DNA Assembly Master Mix. The PCR primers used are listed in Supplementary Data 7.

**Transformation**. The constructed binary vectors were introduced into *Agrobacterium tumefaciens* (strain GV3101). For a generation of *A. thaliana* stable lines, the floral dip method was used to introduce the constructed vectors into *A. thaliana* plants[32]. In the T₁ generation, seedlings carrying each transgene were identified based on hygromycin or kanamycin resistance. For transient expression in *N. benthamiana* leaves, an agroinfiltration-based transient gene expression method was used with minor modifications[33]. Three days after infiltration, leaves were harvested for protein extraction or observation.

**Imaging system and analysis**. Samples were observed under a FV1200 inverted confocal microscope equipped with a GaAsP detector (Olympus; http://www.olympus-lifescience.com/ja/). A 60× water or 100× oil immersion objective lens with a 405 nm laser was used for DAPI, a 473 nm laser for EGFP or Alexa fluor 488, a 561 nm laser for mRuby, and a 635 nm laser for Alexa fluor 647. For live imaging, 7-day-old seedlings pre-incubated on MS plates were transferred to 35 mm glass-based dishes (IWAKI) and covered with a piece of MS solid medium. For the ratio calculation using the two-component system, we acquired Z-stack images and compiled a projection into one layer using the Z Project (Sum Slices) of FIJI[34]. For the ratio calculation in Figs. 4c, d, and 5c, EGFP and mRuby intensities were measured using FIJI for manually selected regions where H2B-mRuby was localized, and the raw data were imported into R to calculate the ratio. In Figs. 2e, 3d, g, and Supplementary Fig. 3b, nuclear region selection (i.e., nucleus segmentation), intensity integration, and ratio calculations were performed semi-automatically using an in-house Python code[35] using the scikit-image library (https://scikit-image.org/). The employed segmentation algorithm takes ideas from the algorithm reported[36], especially the idea of using circular filter banks for detecting cell nucleus positions.

**Immunostaining**. Roots from 7-day-old seedlings were fixed by immersion for 30 min at 4 °C in 4% formaldehyde and phosphate-buffered saline (PBS), then washed three times in PBS and chopped to isolate nuclei in chopping buffer [15 mM Tris-HCl (pH 7.5), 2 mM EDTA (pH 8.0), 0.5 mM spermine, 80 mM KCl, 20 mM NaCl, and 0.1% Triton X-100]. Nuclei were passed through Miracloth (Merck Millipore), and the filtered nuclei suspension was spread on an APS-coated slide glass (MATSUNAMI). The slide glasses were incubated overnight at 4 °C with primary antibodies at the following dilutions: 1:100 for GFP (11814460001; Roche) and 1:100 for RNAPII Ser2P (ab193468; Abcam) in PBS with 4% bovine serum albumin. The slides were washed in PBS three times for 5 min each. Slides were incubated for 1 h at 37 °C with conjugated secondary antibodies used at the following dilutions: 1:1000 for Alexa fluor 488 (ab150113; Thermo Fisher Scientific) and 1:1000 for Alexa fluor 647 (ab150079; Thermo Fisher Scientific). After three washes in PBS for 5 min each, the slides were incubated with 1 μg/mL DAPI in PBS for 5 min, washed for 5 min in PBS, and mounted in VECTASHIELD Mounting Medium (H-1000; Vector Laboratories, Inc.).

**Western blotting analysis**. Protein extracts were mixed with sodium dodecyl sulfate-polyacrylamide gel electrophoresis (SDS-PAGE) sample buffer and denatured at 95 °C for 5 min. After brief centrifugation, denatured protein samples were subjected to 8 or 12% SDS-PAGE and blotted onto PVDF membranes. Primary and secondary antibodies were used at the following dilutions: 1:10,000 for GFP (ab290; Abcam); 1:10,000 for β-tubulin (MAB3408; Sigma-Aldrich); 1:5000 for RNAPII Ser2P (MABI0602; MAB Institute); 1:10,000 for RFP (R10367; Thermo Fisher Scientific); 1:10,000 for anti-IgG (rabbit) pAb-HRP conjugate (458; MBL); and 1:10,000 for anti-IgG (mouse) HRP conjugate (W402B; Promega). Immunoreactive bands were visualized by ImmunoStar® LD (Wako). The molecular weights were estimated with the ExPASy: SIB bioinformatics resource portal[37].

**Chemical treatment**. For the treatment with flavopiridol (ChemScene), 7-day-old seedlings pre-incubated on MS plates were transferred to control MS solid or liquid medium supplemented with 0.1% EtOH (Mock) and MS solid or liquid medium supplemented with 10 μM flavopiridol. The ratio was calculated in cortex cells in the root transition zone using the Python code.

**Immunoprecipitation**. Nuclear proteins were isolated from 10-day-old seedlings as described previously[38] with some modifications. Sonication was conducted using a BIORUPTOR® UCD-250HSA (COSMO BIO CO., LTD.) using power mode H and on/off cycle of 30 s/60 s for a total duration of 12 min on ice in sonication buffer [500 mM LiCl, 0.7% sodium deoxycholate, 1% NP-40, 50 mM HEPES-KOH (pH 7.6), 1 mM EDTA (pH 8.0), working concentration of cOmplete, EDTA-free (Roche), 1 mM Pefablock® SC (Roche), and 5 mM NaF]. The following antibody and magnet beads were used: mouse anti-GFP (11814460001; Roche), mouse anti-RNAPII Ser2P (MABI0602; MAB Institute), and Dynabeads™ M-280 sheep anti-mouse IgG (11201D; Thermo Fisher Scientific).

**ChIP-seq analysis**. Libraries were pooled and 75 bp single-read sequences were obtained using a NextSeq 500 sequencer (Illumina). Genome-wide localization patterns of Ser2P-mintbody and RNAPII Ser2P were analyzed using *pUBQ10: Ser2P-mintbody* transgenic plants and control transgenic plants carrying p35S: EGFP-3xFLAG. Ten-day-old seedlings underwent ChIP-seq analysis using anti-GFP mouse antibody and RNAPII Ser2P antibody as described above.

Quality-filtered reads were mapped onto the *Arabidopsis* reference genome TAIR10 using Bowtie2[39,40] in a mode to report only uniquely mapped reads. The resulting SAM files were converted to a sorted BAM file using SAMtools[41], then converted to BED files using BEDTools[42]. The "slop" function of BEDTools was used to extend the 5′ end of ChIP-seq reads toward the 3′ direction to fit the average insertion size (250 bp) of the sequenced libraries. Ser2P-mintbody and RNAPII Ser2P-distributed sites were detected using Model-based Analysis for ChIP-seq (MACS2[43]) with reads from the anti-GFP (*p35S:EGFP-3xFLAG*) sample used as the control ($q < 0.05$). For visualization, TDF files were created using igvtools from BAM files and visualized with Integrative Genome Viewer[44]. The ngs.plot.r program[45] was used to visualize the distribution around gene bodies or detected peaks.

**Statistics and reproducibility**. The number of samples in experiments is shown in the figures. Comparisons between groups were determined using the Wilcoxon rank-sum two-sample test, shown in Fig. 3d, the two-tailed unpaired two-sample *t* test, shown in Fig. 3g, and the Bonferroni-adjusted Wilcoxon rank-sum test, shown in Fig. 4e. All *p* values are shown in the graphs in Fig. 3d, g, and *p* values less than 0.05 are shown in Fig. 4e. All source data and *p* values are available as Supplementary Data 1–6. The analyses were performed on at least duplicated biological replicates.

**Reporting summary**. Further information on research design is available in the Nature Research Reporting Summary linked to this article.

## Data availability
Data generated or analyzed during this study are included in this published article (and its supplementary information files). Unprocessed blots can be seen in Supplementary Figs. 4–6. Supplementary Data 1 contains source data corresponding to Fig. 2e, Supplementary Data 2 contains source data corresponding to the Fig. 3d, Supplementary Data 3 contains source data corresponding to Fig. 3g, Supplementary Data 4 contains source data corresponding to Fig. 4c–e, Supplementary Data 5 contains source data corresponding to the Fig. 5c, and Supplementary Data 6 contains source data corresponding to the Supplementary Fig. 3. The PCR primers used in plasmid construction are listed in Supplementary Data 7. ChIP-seq data that support the findings of this study have been deposited in the DNA Data Bank of Japan (accession No. DRA011114).

## Code availability
The code for the fluorescence intensity ratio calculations of this study has been deposited in Zenodo (https://doi.org/10.5281/zenodo.4628571).

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

## Acknowledgements

We thank Ms. Emi Kato and Ms. Sayoko Mibu for their technical assistance. We thank Robert McKenzie, Ph.D., from Edanz Group (https://en-author-services.edanz.com/ac) for editing a draft of the paper. M.K.S. is supported by MEXT/JSPS KAKENHI grant No. 20K15836. In addition, T. Sakamoto is supported by KAKENHI grant Nos. 19K06748 and 20H05425, H.K. by 18H05527 and S.M. by 19H03259, 20H03297, and 20H05911. S.M. is also supported by Novartis Foundation, Mitsubishi Foundation. S.M. and Y.S. are supported by JST, CREST grant No. JPMJCR20S6.

## Author contributions

M.K.S., T. Sakamoto, and S.M.: conception and design of the experiments and interpretation of data; M.K.S., H.K., and S.M.: project initiation; M.K.S., T.Y., and M.Y.: acquisition of data; M.K.S. and S.K.: analysis of data; M.K.S.: interpretation of data and drafting the paper; T. Sakamoto, S.M., and S.K.: revision of the paper; M.K.S., S.F., N.Y., S.U., Y.S., and H.K.: mintbody and transgene constructions; T. Suzuki: sequencing. The paper was written based on inputs from all authors.

## Competing interests
The authors declare no competing interests.
