## [Peer Review File · Communications Biology]

Reviewers' Comments:

Reviewer #1:

Remarks to the Author:

The manuscript by Shibuta et al., entitled "A live imaging system to analyze spatiotemporal dynamics of modification of RNA polymerase II in plants" is indeed very interesting.

The authors generated a mintbody hooked to a EGFP reporter which contains both heavy and light chains fragment (scFv). The authors subsequently performed ChIP-seq analyses and demonstrated that both the mintbody and the anti-Ser2P antibody had rather similar patterns targeting the transcriptional end site. Common peaks were identified for Ser2P-mintbody versus Ser2P (9572), and Ser2P only targets (4264) versus Ser2P-mintbody only (336). An EGFP control is used as control in SupFig 2b, which provides certainty over the quality of the data.

Next, the authors generated co-expression cassettes using an internal element termed IntF2A, which self-excise at high efficiency. They further showed that control plants cleaved well under standard conditions.

Subsequently, the authors generated a reporter containing a pRPS5a:Ser2P-mintbody-IntF2A-H2B-mRuby, and pRPS5a:EGFP-IntF2A-H2B-mRuby. They used the drug Flavopiridol that blocks kinase activity, and showed that subsequent washes relocalized the Ser2P-mintbody in the nucleus after washing.

The authors next showed that transcriptional activity was significantly reduced during the M phase, which is likely the highlight of the paper. They again showed that both the pRPS5a:EGFP-intF2A-H2B-mRuby and the pRPS5a:Ser2P-mintbody-IntF2A-H2B-mRuby had similar patterns of localisation during M phase. Interestingly, this suggests that a drop of transcription happens from prometaphase to telophase.

All together, I would certainly recommend the paper for publication in its current form. All statistics and extremely well described, and the ChIP data is very well documented.

Reviewer #2:

Remarks to the Author:

The study by Shibuta et al reports on the generation, verification and use of a novel live reporter system for monitoring global transcriptional levels in planta. The system makes use of a genetically encoded nanobody (Minbody) against phosphor-Serine 2 (Ser2P) Pol II isoform. The authors present convincing evidence by chromatin immunoprecipitation that the Mintbody binds in a similar way than the endogenous Ser2P-RNA PolII. Further, they developed a strategy to co-express an internal standard (H2B-mRuby) which is cleaved post-translationally from the Ser2P-mintbody containing protein fusion. This elegant and functional approach provides a ratiometric method for quantifying the fraction of Ser2P RNA Pol II isoforms. As proof of concept, an analysis of Ser2P levels in different root tissue and during the cell cycle in root cells is provided. The manuscript is well written and structured, the work is well explained and documented, the figures are very informative. The conclusions are very well supported by the experiments.

I have only one reserve on experiments presented Figure 1b and 1c, which can be easily solved given the very strong evidence provided by ChIP seq in Fig 1d-e and Figure S2, and some minor text issues listed at the end.

The colocalisation of mintbody-GFP at RNAPolII Ser2P regions is assessed in two different ways: using co-immunostaining and using co-immunoprecipitation.

While the first experiment allows to conclude similar distribution of the signals in euchromatin, it cannot assess colocalisation at Ser2P-enriched foci (transcription factories). An explanation for the lack of full colocalisation is provided lines 104-105 suggesting a pool of free mintbody-GFP. This is, however, unlikely the explanation as one would expect a diffuse signal and not foci of enriched

signal as it is currently the case. Instead, it is likely that mintbody-GFP and the anti-Ser2P antibody compete for the same target. Hence, the conclusion can only be made for a similar distribution in euchromatin but cannot assess colocalisation at transcription factories.

The co-immunoprecipitation experiment is in theory the most decisive one to test the fraction of Ser2P effectively bound by the Mintbody line. Unfortunately, the gel currently presented Figure 1c is not entirely convincing. The Ser2P band is really weak. The full gel representation in supplemental figure 4 is not mentioned in the text together with Fig1c (hence I initially missed it and thought it was not provided) and replicate experiments are not shown. In addition, a sensible analysis would be to quantify the fraction of Ser2P bound by Mintbody compared to Ser5P and non-phosphorylated CTD. These fractions could be expressed relative to total PolII, using different antibodies (against Ser2P-CTD, Ser5P-CTD, NP-CTD and total PolII, antibody exist against the RPB1 subunit)

On the other hand, this experiment is obsolete with the ChIPseq experiment. I thus suggest to either improve the co-IP or to remove it as it raises question due to its poor quality and current presentation rather than providing certainty.

Minor issues

L193-104 "Ser2P and was exclusively localized with DAPI-stained heterochromatin (Fig. 1b)." The authors mean "excluded from heterochromatin" or "exclusively localized in euchromatin" ??

L104-105 - add also possible explanation involving competition of the anti-GFP and the mintbody for the same target

L118 - remove extremely. It is either similar or it is not.

L132 "and was an effective tool for monitoring the dynamics of Ser2P" is not yet demonstrated at that stage of reading (it comes next section) - suggestion to remove.

L186 "gradually diffused from the nucleus" -> diffused away from the nucleus?

L190 "the nuclear-biased distribution pattern" -> reaffirming the preferential nuclear distribution pattern, or preferential distribution in the nucleus

L199 "a tissue-specific transcription-level atlas", not sure that atlas is the proper term here. In developmental biology it specifically refers to gene expression (type, level) associated with specific cell or tissue type in an organ and/or throughout a developmental period. Also an atlas of transcription level is not clear. Perhaps better "To generate a tissue-specific map of transcriptional activity" ?

L241-242 "In addition, we detected transcription at the individual-gene level (Fig. 1d-g).", this wrongly suggest that the Ser2P-Mintbody detects a live transcription process at the gene level, referring to the imaging experiments largely documented in this manuscript. Please correct for "in addition, we show that the mintbody construct can be used to map the enrichment of Ser2P-PolII genome wide in chromatin immunoprecipitation experiments.

Discussion- it is classically devoid of reference to figures

L249-250 "This lower binding affinity means less disturbance of the protein function, and enables normal growth without aberrant phenotypes" what do you mean with less disturbance? Less than what? Normal antibodies are anyway not used in live imaging, but in immunostaining on fixed tissue. Also I am not sure that the affinity would be linked to the disturbance of the protein function. Putative adverse effect of a nanobody (or any heterologous binding protein) may be more expected in relation to the steric hindrance or structural obstacle to binding other partners. Hence here, simply stating that the nanobody expression did not show any obvious adverse effect on development may be sufficient.

L251-252 "transcription level...was relatively varied", you mean "variable"?

Title supplemental figure 2 "SFig. 2: ngs plots for evaluation of the binding capacity of Ser2P-mintbody with Ser2P." What means ngs plot? Please write the full name. If ngs means "next generation sequencing", then "ngs plots" is incorrect. In addition, these plots do not allow to evaluate the "binding capacity" of mintbody "to" Ser2P. These are not co-IP experiments, rather, separate experiments. Correct for "xx plots showing that mintbody and RNA PolII-Ser2P have a similar distribution profile and localize to common loci

Point-by-point responses to the reviewers' comments

To Reviewer#1

Comment: The manuscript by Shibuta et al., entitled “A live imaging system to analyze spatiotemporal dynamics of modification of RNA polymerase II in plants” is indeed very interesting. The authors generated a mintbody hooked to a EGFP reporter which contains both heavy and light chains fragment (scFv). The authors subsequently performed ChIP-seq analyses and demonstrated that both the mintbody and the anti-Ser2P antibody had rather similar patterns targeting the transcriptional end site. Common peaks were identified for Ser2P-mintbody versus Ser2P (9572), and Ser2P only targets (4264) versus Ser2P-mintbody only (336). An EGFP control is used as control in SupFig 2b, which provides certainty over the quality of the data.

Next, the authors generated co-expression cassettes using an internal element termed IntF2A, which self-excise at high efficiency. They further showed that control plants cleaved well under standard conditions. Subsequently, the authors generated a reporter containing a pRPS5a:Ser2P-mintbody-IntF2A-H2B-mRuby, and pRPS5a:EGFP-IntF2A-H2B-mRuby. They used the drug Flavopiridol that blocks kinase activity, and showed that subsequent washes relocalized the Ser2P-mintbody in the nucleus after washing. The authors next showed that transcriptional activity was significantly reduced during the M phase, which is likely the highlight of the paper. They again showed that both the pRPS5a:EGFP-intF2A-H2B-mRuby and the pRPS5a:Ser2P-mintbody-IntF2A-H2B-mRuby had similar patterns of localisation during M phase. Interestingly, this suggests that a drop of transcription happens from prometaphase to telophase. All together, I would certainly recommend the paper for publication in its current form. All statistics and extremely well described, and the ChIP data is very well documented.

Response: Thank you for your enthusiastic evaluation of our work.

To Reviewer#2

Comment: The study by Shibuta et al reports on the generation, verification and use of a novel live reporter system for monitoring global transcriptional levels in planta. The system makes use of a genetically encoded nanobody (Minbody) against phosphor-Serine 2 (Ser2P) Pol II isoform. The authors present convincing evidence by chromatin immunoprecipitation that the Mintbody binds in a similar way than the endogenous Ser2P-RNA PolII. Further, they developed a strategy to co-express an internal standard (H2B-mRuby) which is cleaved post-translationally from the Ser2P-mintbody containing protein fusion. This elegant and functional approach provides a ratiometric

method for quantifying the fraction of Ser2P RNA Pol II isoforms. As proof of concept, an analysis of Ser2P levels in different root tissue and during the cell cycle in root cells is provided. The manuscript is well written and structured, the work is well explained and documented, the figures are very informative. The conclusions are very well supported by the experiments.

Response: Thank you for your favorable evaluation that the findings are of great interest. We modified the manuscript in accordance with your suggestions.

Comment 1: While the first experiment allows to conclude similar distribution of the signals in euchromatin, it cannot assess colocalisation at Ser2P-enriched foci (transcription factories). An explanation for the lack of full colocalisation is provided lines 104-105 suggesting a pool of free mintbody-GFP. This is, however, unlikely the explanation as one would expect a diffuse signal and not foci of enriched signal as it is currently the case. Instead, it is likely that mintbody-GFP and the anti-Ser2P antibody compete for the same target. Hence, the conclusion can only be made for a similar distribution in euchromatin but cannot assess colocalisation at transcription factories.

Response: We agree with the reviewer's comment. In accordance with your suggestion, we have added the following sentence to L102–104 of the revised manuscript: “, and that anti-GFP antibody for Ser2P-mintbody and the anti-Ser2P antibody would compete for the same target,”

Comment 2: The co-immunoprecipitation experiment is in theory the most decisive one to test the fraction of Ser2P effectively bound by the Mintbody line. Unfortunately, the gel currently presented Figure 1c is not entirely convincing. The Ser2P band is really weak. The full gel representation in supplemental figure 4 is not mentioned in the text together with Fig1c (hence I initially missed it and thought it was not provided) and replicate experiments are not shown. In addition, a sensible analysis would be to quantify the fraction of Ser2P bound by Mintbody compared to Ser5P and non-phosphorylated CTD. These fractions could be expressed relative to total PolII, using different antibodies (against Ser2P-CTD, Ser5P-CTD, NP-CTD and total PolII, antibody exist against the RPB1 subunit). On the other hand, this experiment is obsolete with the ChIPseq experiment. I thus suggest to either improve the co-IP or to remove it as it raises question due to its poor quality and current presentation rather than providing certainty.

Response: We appreciate the reviewer's comment on this point. As suggested, the Ser2P band in the immunoprecipitated (IP) lane was weak. We performed the

experiment many times, but always got the weak band. Thus, in accordance with your latter comment, we have removed the co-immunoprecipitation experiment (Fig. 1c) from the revised manuscript.

Comment 3: L103-104 “Ser2P and was exclusively localized with DAPI-stained heterochromatin (Fig. 1b).” The authors mean “excluded from heterochromatin” or “exclusively localized in euchromatin” ??

Response: Thank you for your comment. We mean “excluded from heterochromatin”. We corrected the wording in the revised manuscript on L100-101.

Comment 4: L104-105 - add also possible explanation involving competition of the anti-GFP and the mintbody for the same target

Response: In accordance with the reviewer’s suggestion, we have added the following sentence on L102-104 in the revised manuscript:

“and that anti-GFP antibody for Ser2P-mintbody and the anti-Ser2P antibody would compete for”

Comment 5: L118 – remove extremely. It is either similar or it is not.

Response: As recommended we removed “extremely” in the revised manuscript.

Comment 6: L132 “and was an effective tool for monitoring the dynamics of Ser2P” is not yet demonstrated at that stage of reading (it comes next section) – suggestion to remove.

Response: Following the reviewer’s suggestion, we have removed the sentence in the revised manuscript.

Comment 7: L186 “gradually diffused from the nucleus “ -> diffused away from the nucleus?

Response: We agree with the reviewer’s suggestion. We have corrected the wording to “diffused away from the nucleus” in the revised manuscript on L179.

Comment 8: L190 “the nuclear-biased distribution pattern” -> reaffirming the preferential nuclear distribution pattern, or preferential distribution in the nucleus

Response: We thank the reviewer for this comment. We have changed the wording to “the preferential distribution in the nucleus” in the revised manuscript on L183.

Comment 9: L199 “ a tissue-specific transcription-level atlas” , not sure that atlas is the proper term here. In developmental biology it specifically refers to gene expression (type, level) associated with specific cell or tissue type in an organ and/or throughout a developmental period . Also an atlas of transcription level is not clear. Perhaps better “To generate a tissue-specific map of transcriptional activity” ?

Response: We thank the reviewer for this comment. Following the reviewer’s suggestion, we have reworded this text to “To generate a tissue-specific map of transcriptional activity” in the revised manuscript on L192-193.

Comment 10: L241-242 “In addition, we detected transcription at the individual-gene level (Fig. 1d–g). “, this wrongly suggest that the Ser2P-Mintbody detects a live transcription process at the gene level, referring to the imaging experiments largely documented in this manuscript. Please correct for “in addition, we show that the mintbody contstruct can be used to map the enrichment of Ser2P-PolII genome wide in chromatin immunoprecipitation experiments.

Response: In accordance with the reviewer’s suggestion, we have changed the sentence to “In addition, we show that the Ser2P-mintbody construct can be used to map the enrichment of RNAPII Ser2P genome wide in ChIP-seq analysis” in the revised manuscript on L234-235.

Comment 11: Discussion- it is classically devoid of reference to figures

Response: We thank the reviewer for this comment. Accordingly, we have removed all references to figures in the Discussion in the revised manuscript.

Comment 12: L249-250 “This lower binding affinity means less disturbance of the protein function, and enables normal growth without aberrant phenotypes” what do you mean with less disturbance? Less than what? Normal antibodies are anyway not used in live imaging, but in immunostaining on fixed tissue. Also I am not sure that the affinity would be linked to the disturbance of the protein function. Putative adverse effect of a nanobody (or any heterologous binding protein) may be more expected in relation to the steric hindrance or structural obstacle to binding other partners. Hence here, simply stating that the nanobody expression did not show any obvious adverse effect on development may be sufficient.

Response: We thank the reviewer for these comments. We mean that Ser2P-mintbody shows less disturbance than the normal antibody. However, as the reviewer mentioned, we cannot compare the binding affinity of these two antibodies because the working conditions are totally different. Therefore, we have removed the sentence in the revised manuscript.

In our study using Ser2P-mintbody, transgenic lines that expressed Ser2P-mintbody did not show an aberrant phenotype. Therefore, we have revised the relevant text in the revised manuscript on L235-240 as follows:

“The mintbody showed a lower peak and fewer target genes than those for the anti-Ser2P antibody, reflecting the moderate affinity of Ser2P-mintbody for the targets (Fig. 1d–g). This at least partially reflects its structure; mintbodies are monovalent, whereas normal antibodies are bivalent. In fact, our generated lines expressing Ser2P-mintbody in this study normally grew without aberrant phenotypes (Supplementary Fig. 1a).”

Comment 13: L251-252 “transcription level...was relatively varied”, you mean “variable”?

Response: We thank the reviewer for this comment. We indeed meant “variable” and have corrected the wording in the revised manuscript on L242.

Comment 14: Title supplemental figure 2 “SFig. 2: ngs plots for evaluation of the binding capacity of Ser2P-mintbody with Ser2P.” What means ngs plot? Please write the full name. If ngs means “next generation sequencing”, then “ngs plots” is incorrect. In addition, these plots do not allow to evaluate the “binding capacity” of mintbody “to” Ser2P. These are not co-IP experiments, rather, separate experiments. Correct for “xx plots showing that mintbody and RNA PolII-Ser2P have a similar distribution profile and localize to common loci.”

Response: We apologize for the misunderstanding about the ngs plot. We have corrected “ngs plot” to “average profile plot” in the revised manuscript on L571, and changed the title of Supplementary Fig. 2 to “Average profile plot showing that Ser2P-mintbody and RNAPII Ser2P have a similar distribution profile and localize to common loci”.

Reviewers' Comments:

Reviewer #1:

Remarks to the Author:

The manuscript by Shibuta et al is a much improved version of the former manuscript, and should be published in its current form. While the structure of the text have been significantly improved, particularly with the addition of the uncropped gel at the end, it is clearly publishable in its current form. I would thus strongly recommend it as is. Congratulation to the authors for their excellent work.

Reviewer #2:

Remarks to the Author:

The authors have responded to all my comments satisfactorily.